# Synthesis and Degradation of Poly(ADP-ribose) in Zebrafish Brain Exposed to Aluminum

**DOI:** 10.3390/ijms24108766

**Published:** 2023-05-15

**Authors:** Anna Rita Bianchi, Alessandra La Pietra, Valeria Guerretti, Anna De Maio, Teresa Capriello, Ida Ferrandino

**Affiliations:** Department of Biology, University of Naples Federico II, Via Cinthia 21, 80126 Naples, Italy; annarita.bianchi@unina.it (A.R.B.); alessandra.lapietra@unina.it (A.L.P.); valeria.guerretti@studenti.unina.it (V.G.); teresa.capriello@unina.it (T.C.)

**Keywords:** zebrafish, brain, aluminum, PARP, PARG, neurodegeneration, poly(ADP-ribose)

## Abstract

Poly(ADPribosyl)ation is a post-translational protein modification, catalyzed by poly(ADP-ribose) polymerase (PARPs) enzymes, responsible for ADP-ribose polymer synthesis (PAR) from NAD^+^. PAR turnover is assured by poly(ADPR) glycohydrolase (PARGs) enzymes. In our previous study, the altered histology of zebrafish brain tissue, resulting in demyelination and neurodegeneration also with poly(ADPribosyl)ation hyperactivation, was demonstrated after aluminum (Al) exposure for 10 and 15 days. On the basis of this evidence, the aim of the present research was to study the synthesis and degradation of poly(ADP-ribose) in the brain of adult zebrafish exposed to 11 mg/L of Al for 10, 15, and 20 days. For this reason, PARP and PARG expression analyses were carried out, and ADPR polymers were synthesized and digested. The data showed the presence of different PARP isoforms, among which a human PARP1 counterpart was also expressed. Moreover, the highest PARP and PARG activity levels, responsible for the PAR production and its degradation, respectively, were measured after 10 and 15 days of exposure. We suppose that PARP activation is related to DNA damage induced by Al, while PARG activation is needed to avoid PAR accumulation, which is known to inhibit PARP and promote parthanatos. On the contrary, PARP activity decrease at longer exposure times suggests that neuronal cells could adopt the stratagem of reducing polymer synthesis to avoid energy expenditure and allow cell survival.

## 1. Introduction

Poly(ADPribosyl)ation is a post-translational modification consisting of the attachment of ADP-ribose (ADPr) units to acidic or basic amino acid residues of target nuclear proteins [1]. The ADPR polymer (PAR) produced by this reaction is synthesized by enzymes known as poly(ADP-ribose) polymerases (PARPs) from NAD^+^, and it can differ in size (number of ADPr units) and shape (linear or branched) [2]. The turnover of poly(ADPR) is ensured by poly(ADP-ribose) glycohydrolase (PARG) enzymes, which degrade it by releasing ADPr molecules [3].

To date, 17 PARP isoforms have been identified, but only one PARG gene [4]. Members of the PARP superfamily have a highly conserved catalytic domain [5] but distinct structures, functions, and localizations [6]. These enzymes have been divided into six groups on the basis of domain architecture (PARP1 subgroup, vault PARP, tankyrase, CCCH-PARP, macro-PARP, and others) and three categories on the basis of enzymatic activity (poly, mono, and inactive) [7]. The most characterized member of this family is the nuclear PARP1, also known as ADP-ribosyltransferase 1 (ARTD1), according to a recently proposed nomenclature [8]. PARP and PAR have been implicated in many cellular processes [4].

As in animals, and in plants, PARP1 is involved in the repair and maintenance of DNA integrity and transcriptional regulation, modulation of chromatin structure, and cell death [9,10].

PAR is known to bind to acceptor proteins through covalent and non-covalent interactions. The covalent binding of PAR influences the structure and modulates the function of the modified protein due to the high density of negative charges on the polymer [11]. Two examples are the self-modification of PARP, which produces an inactivation of the enzyme, and the hetero-modification of histones, which is involved in chromatin decondensation to facilitate access to damaged DNA [12].

Non-covalent interactions between poly(ADPR) and specific protein acceptors play a role in DNA damage response and regulation of protein stability [13].

The chromatin remodeling enzyme ALC1 (Amplified in Liver Cancer 1, also known as CHD1L) is a DNA damage response protein, as it is rapidly recruited to DNA damage sites in a poly(ADP-ribose)-dependent manner [14].

PARP1 is considered a “marker” of DNA damage, as genotoxic stress leads to PARP1 activation and the consequent decrease in intracellular NAD^+^ and ATP levels [15].

In detail, when genotoxic DNA damage is mild, PARP is activated and promotes genomic material repair and cell survival [16]. On the contrary, prolonged PARP hyperactivation due to massive DNA damage can lead the cell to death by apoptosis or necrosis [17].

PARP activation has also been correlated to severe water stress resistance [18], pollution [19], steroid-regulated physiology disruption, exposure to heavy metals, ionizing radiation [20,21,22], and aging when a higher production of radical species occurs [23].

In addition to physiological events, poly(ADPribosyl)ation is also involved in various pathologies, such as cancer; diabetes; and autoimmune, cardiovascular, and neurodegenerative diseases [24,25,26,27,28].

Aging is the dominant risk factor associated with neurodegenerative diseases, linked to oxidative stress, which is responsible for the dysfunction or death of neuronal cells [29,30,31]. 

During normal aging, the brain is an organ concentrating metal ions with alteration of homeostasis that seems to play a role in a variety of age-related neurodegenerative diseases [32,33].

In recent years, living beings have been exposed to biologically reactive aluminum, a trivalent cation that is highly neurotoxic and whose effects have been linked to multiple neurological disorders of the human central nervous system (CNS).

The constant accumulation and compartmentalization of aluminum (Al) in the aging human CNS have been involved in several neuropathological processes and pathological mechanisms, such as amyloidogenesis, alteration of innate immunity, pro-inflammatory signaling, neural degeneration, and alterations in the expression of genetic information, which is essential for the brain [34].

These neurochemical and molecular genetic changes have been related to deficits in behavior, cognition, and memory in patients with neurological and/or neurodevelopmental conditions [35,36].

Aluminum alters blood–brain barrier permeability, allowing it to reach the central nervous system [37], and its accumulation can cause Alzheimer’s disease, Parkinson’s disease, and amyotrophic lateral sclerosis [38,39,40,41]. Several studies have linked aluminum accumulation to neurodegenerative pathogenesis, on the basis of evidence that aluminum could exacerbate oxidative events, activate the production of reactive oxygen species, and induce oxidative stress [42,43,44].

Our previous studies showed an increase in apoptosis, oxidative stress, and genotoxicity in zebrafish larvae exposed to different aluminum concentrations in the first days of development [45]. Moreover, in adult zebrafish, Al induced the alteration of swimming ability and behavior but also the histology and the oxidative stress of their brain [46].

Recently, we also showed PARP hyperactivation as an immediate response to oxidative DNA damage in the brain of zebrafish exposed to 11 mg/L Al, which is the concentration revealed in polluted sites [47].

On the basis of this evidence, the aim of the present research was to study, for the first time, the complete poly(ADPribosyl)ation system in the brain of zebrafish. Zebrafish is an organism widely used to study human diseases [48] and the effects of various pollutants [45,49,50] due to its characterized and conserved genome [51], with 70% similarity to their mammalian orthologues [52].

In detail, in this study, we carried out the analysis of PARP and PARG expression in order to demonstrate the presence of both enzymes (PARPs) involved in poly(ADPR) synthesis and enzymes (PARG) responsible for its degradation. 

We also purified poly(ADPR) and subsequently analyzed the polymer and its degradation product on both high-resolution sequence gels and thin-layer chromatography (TLC).

## 2. Results

### 2.1. Identification of PARP Enzymes by Immunochemical Analysis

Control (Ctrl) and brain homogenates exposed for 10 (T10), 15 (T15), and 20 days (T20) to Al were analyzed by 12% polyacrylamide gel in 0.1% sodium dodecyl sulphate (SDS) (Figure 1a) and immunoblotting with anti-PARP (Figure 1b), anti-PARP1 N20 (Figure 1c), and anti-PAR (Figure 1d) antibodies.

Electrophoretic protein patterns showed no significant qualitative and quantitative differences (Figure 1a).

The immunoblotting with anti-PARP evidenced several immunoreactive signals, corresponding to proteins with molecular weight between 30 and 113 kDa (Figure 1b). Only one signal of 113 kDa was observed following incubation with anti-PARP1 N20, able to recognize the zinc finger N-terminal domain of PARP1 (Figure 1c). Finally, different immunopositive signals to anti-PAR, corresponding to covalent protein acceptors of poly(ADPR), were detected in samples exposed to Al for 10 and 15 days (Figure 1d). Figure 1e shows the Western blot normalized to β-actin.

Densitometric analysis of the 113 kDa band in Figure 1c showed that there was no difference in the signal intensities measured in the control and in the samples exposed to Al (*p* < 0.05) (Figure 1f).

### 2.2. PARG Expression

The presence of the PARG enzyme, which is able to hydrolyze poly(ADP-ribose), was investigated by Western blotting. The electrophoretic analysis confirmed that there were no significant qualitative and quantitative differences in the proteic pattern (Figure 2a). Two immunoreactive signals corresponding to proteins of 68 kDa and 87 kDa were recognized by the anti-PARG antibody (Figure 2b). Western blot normalized to β-actin is shown in Figure 2c.

No difference in intensity (*p* < 0.05) was measured in both signals following densitometric analysis (Figure 2d).

### 2.3. PARG Assay

Poly(ADPR) glycohydrolase activity was assayed on [^32^P]ADP-ribosylated proteins produced from control, T10, T15, and T20 brain homogenates by PARP assay in order to demonstrate PAR turnover.

In Table 1, PARP and PARG activity levels are reported in order to demonstrate PAR synthesis and its degradation. The highest PARP and PARG activities were measured in brain homogenate exposed to Al for 10 and 15 days. At longer exposure times (20 days), instead, both enzyme activities returned to levels comparable to those detected in the control (Table 1).

Finally, there was a complete degradation of the PARP reaction product in both the control and in all samples exposed to Al (Table 1).

### 2.4. TLC and PAGE of poly(ADPR) Synthesis and Degradation

PAR and its degradation product were analyzed by high-resolution sequence gels (Figure 3).

PAGE of [^32^P]-labeled poly(ADPR) showed short ADPR oligomers (of about six units) in control and T20 samples and polymers of about 30–35 units in T10 and T15 (Figure 3).

Furthermore, the poly(ADPR) synthesized in the brain homogenate of T15 was converted to ADP-ribose units under PARG assay conditions (Figure 3, line T15*).

The PAR synthesized in T15 samples was also analyzed by thin-layer chromatography (Figure 4). The results indicated that the protein-free polymer was completely degraded to ADPR, as confirmed by the disappearance of the radioactive signal in the TLC loading point and the simultaneous appearance of a signal that co-migrates with standard ADP-ribose (Figure 4).

## 3. Discussion

In this paper, we focused on the poly(ADPribosyl)ation reaction, catalyzed by PARP enzymes, which are activated following ROS-induced genotoxic damage [53]. ROS are involved in neurodegenerative diseases, ultimately characterized by the death of neural cells [54]. Several studies demonstrated that poly(ADPribosyl)ation activation is one of the possible modulators of neurodegeneration [55], as poly(ADPR) is a signaling molecule for several forms of cell death [56].

Although increased aluminum levels in the environment are known to have negative consequences for the neurological health of living beings [57], there is growing evidence that even low concentrations can cause negative effects [42]. The mechanism by which this compound promotes the onset and development of neurodegenerative diseases is probably due to an acceleration of intrinsic undesirable events that already occur in the aging brain [58]. The most deleterious of these is the progressive increase in inflammatory events associated with ROS increase during aging [59].

In our previous studies, we demonstrated altered histology of zebrafish brain tissue, resulting in demyelination and neurodegeneration in the first period of Al exposure (10 and 15 days). In addition, an hyperactivation of poly(ADPribosyl)ation was also detected in response to oxidative damage to genomic material at the same time of exposure [47].

In the light of these data, in the present research, a complete characterization of the poly(ADPribosyl)ation system in the zebrafish brain, including both the poly(ADPR) synthesis, catalyzed by PARP enzymes, and its degradation by PARGs, was conducted.

Expression protein analysis by anti-PARP antibody, able to recognize the high conserved catalytic site of these enzymes, revealed the presence of different PARP isoforms, having different molecular weights between 30 and 113 kDa (Figure 1b). These data confirm what is reported in UniProt about zebrafish PARPs [60].

The 113 kDa protein could correspond to human PARP1 (hPARP1) [61]. This hypothesis was also confirmed by evidence that zebrafish PARP (zPARP) as hPARP1 possesses at the N-terminal end three zinc-finger domains, which were recognized by the anti-PARP N20 antibody (Figure 1c). Finally, zPARP as hPARP1 was also shown to be a covalent acceptor of poly(ADPR) (Figure 1d). The proteins of about 70 kDa and 60 kDa could correspond to hPARP2 and hPARP3, respectively. As PARP1, mammalian PARP2 is also located in the nucleus and binds DNA, sensing single- and double-strand breaks. Activation of both PARPs leads to the recruitment of DNA repair proteins, histone release, and chromatin decondensation [62,63]. Down- or upregulation of PARP activity confers protection against harmful effects of several forms of abiotic stress, such as ionizing radiation, low temperatures, pH, dehydration, and high light [18,20,64,65,66,67]. PARP3 is a recently characterized member of the PARP family. Although it shows high structural similarities with PARP1 and PARP2, relatively little is known about its cellular properties in vivo [68]. Studies conducted on its characterization in human and mouse models report PARP3 as a newcomer in genome integrity and as a critical player in mitotic spindle stabilization and telomere integrity [68].

As a covalent acceptor of poly(ADPR), zPARP is auto-modified with the polymer of about 30–35 ADPR units after 10 and 15 days of Al exposure (Figure 3 and Figure 4), exactly when PARP activity and PAR synthesis increased. These two events represent indirect evidence of genotoxic damage induced for short times in zebrafish brains, with PARP being a sensor of DNA damage.

On the contrary, we hypothesize that the reduction of polymer length and synthesis, together with the decrease in PARP activity in both control samples and brain exposed to Al for 20 days, could allow the zebrafish brain to adapt to survive longer exposure times. PARP is known to be one of the main consumers of intracellular energy, so reducing its activity would maintain the energy homeostasis essential for the continuation of all vital activities in the organism [69].

Finally, two poly(ADP-ribose) glycohydrolases with different molecular weights (68 and 87 kDa) were identified (Figure 2b), demonstrating the presence of enzymes involved in PAR turnover in the zebrafish brain (Figure 3 and Figure 4). The highest PARG activity was measured after 10 and 15 days of exposure to Al (Table 1), when more polymer is synthesized and with greater length (Figure 3).

We suppose that an equilibrium between the synthesis and degradation of poly(ADPR) at 10 and 15 days to Al exposure occurs. This equilibrium could prevent the accumulation of PAR, which is known to act as a cell death signal for parthanatos, causing a neuronal loss in several neurological diseases [24]. In addition, PAR degradation is necessary to prevent the inhibition of PARP, which at these exposure times is engaged in repairing genotoxic DNA damage [56].

## 4. Materials and Methods

### 4.1. Zebrafish Housing

Adult zebrafish were housed in rectangular tanks under standard conditions as previously described [47] and fed with a commercial diet (TetraMin Tropical Flake Fish^®^) supplemented with *Artemia* sp. Nauplii [70]. All experiments were performed in accordance with the guidelines dictated by European regulations on animal welfare (Directive, 2010/63/EU) and approved by the Italian Ministry of Health (Permit Number: 147/2019-PR). During fish treatment, the water parameters were monitored daily and maintained in the following ranges: temperature 28 °C and pH 7.6.

### 4.2. Treatment Solution

The treatment solution was prepared by dissolving AlCl_3_⋅6H_2_O (Carlo Erba, Cornaredo, Italy) in water (1/3 distilled water, 2/3 tap water), as previously described [45], in order to obtain the final concentration of 11 mg/L Al. Al concentration was already used in the previous study and is linked to that found in polluted waters [46,47].

### 4.3. Assessment of Al Concentration

Three animals were used for each experimental group. Three groups of fish were exposed for 10 (T10), 15 (T15), and 20 (T20) days to 11 mg/L Al separately with daily renewal of solution and monitoring of the water parameters, as previously reported [46,47]. Another group of fish was the control group (Ctrl), exposed only to tank water (1/3 distilled water, 2/3 tap water). The brain was collected after the fish was euthanized with an overdose of MS-222 (Sigma Aldrich, Steinheim am Albuch, Germany). This was performed in triplicate for each experimental group and according to the 3Rs (Replacement, Reduction, and Refinement) principle; in order to maximize the information obtained per animal and thus limit the subsequent use of additional animals, some samples used in this study were taken from organisms already employed in our previous studies [41,46,47].

### 4.4. Homogenates Preparation

Control (Ctrl) and brain samples (4 mg) exposed for 10 (T10), 15 (T15), and 20 days (T20) to 11 mg/L aluminum (Al) from adult zebrafish were homogenized in Buffer A containing 10 mM Tris-HCl (pH 7.5), 1 mM EDTA, 1 mM EGTA (pH 8), 1 mM β-mercaptoethanol, 0.15 mM spermine, 0.75 mM spermidine, 1 mM PMSF, and 2 μg/mL protease inhibitor cocktail, as described in [47]. Protein concentration was determined by the Bradford assay (BioRad, Hercules, CA, USA).

### 4.5. SDS-PAGE and Immunoblotting

Homogenates (20 µg) from all zebrafish brain samples were electrophoresed onto 12% polyacrylamide mini-gel in 0.1% sodium dodecyl sulfate (SDS), as described in Liguori et al. [71]. Staining was in 0.1% Comassie G in 10% acetic acid/30% methanol.

Immunoblotting experiments were performed by electrotransferring proteins onto PVDF membrane (0.2 μm, BioRad) at 200 mA for 2 h at 4 °C. The filter was incubated with these primary antibodies: monoclonal anti-poly(ADP-ribose) polymerase (sc-8007, Santa Cruz Biotechnology, Inc., Dallas, TX, USA, 1:500), polyclonal anti-PARP1 N20 (sc-1561, Santa Cruz, Biotechnology, Inc., 1:1000), monoclonal anti-poly ADP-ribose (PAR) (sc-56198, Santa Cruz, Biotechnology, Inc., 1:500), and monoclonal anti-β-actin (sc-517582, Santa Cruz Biotechnology, Inc., 1:500).

Horseradish peroxidase (HRP)-conjugated anti-mouse secondary antibody (sc-525409, Santa Cruz Biotechnology, Inc., 1:2000) was used to recognize anti-PARP, anti-PAR, and anti-β-actin, while horseradish peroxidase (HRP)-conjugated mouse anti-goat IgG secondary antibody (sc-2354, Santa Cruz Biotechnology, Inc., 1:2000) was utilized for anti-PARP1 N20.

Three stripping procedures were used to remove primary antibodies from the filter, according to Arena et al. [23]. The first stripping removed the anti-PARP to allow incubation with anti-PARP1 N20. Subsequently, this was removed by the second stripping, before proceeding with incubation with anti-PAR. Finally, after the third stripping, the filter was incubated with anti-β-actin.

Immunodetection by enhanced chemiluminescence (ECL, 32106, Thermo Fisher Scientific Inc., Waltham, MA, USA) and quantization by densitometry was conducted by Image Lab 5.2.1 software in a ChemiDoc system (BioRad).

SDS-PAGE and immunoblotting of all zebrafish brain homogenates (20 µg) were carried out using anti-poly (ADP-ribose) glycohydrolase (PARG) (27808-1-AP, Proteintech Group Inc., Rosemont, IL, USA, 1:2000) as primary antibody and horseradish peroxidase (HRP)-conjugated goat anti-rabbit antibody (31460, Thermo Fisher Scientific Inc., 1:2000) as the secondary antibody, as described above. After stripping, the filter was incubated with monoclonal primary anti-β-actin antibodies (sc-517582, Santa Cruz, Biotechnology, Inc., 1:500). As secondary antibody was used, namely, horseradish peroxidase (HRP)-conjugated anti-mouse secondary antibody (sc-525409, Santa Cruz Biotechnology, Inc., 1:2000).

### 4.6. PAGE and TLC of Poly(ADPR) Synthesis

Two aliquots of all homogenates (200 μg of proteins) from zebrafish brain were incubated in a reaction mixture (final volume 500 μL) containing 0.5 M Tris-HCl (pH 7.5), 50 mM MgCl_2_, 10 mM DTT, and 0.5 mM [^32^P]NAD^+^ (100,000 cpm/nmol) for 15 min at 25 °C under PARP assay conditions [22]. The reaction was stopped on ice, and the proteins were precipitated by adding 30% (*w*/*v*) trichloroacetic acid (TCA). After centrifugation at 4000 rpm at 4 °C for 20 min and three washes in absolute ethanol, the precipitates were resuspended in 1 mM EDTA and 10 mM Tris-NaOH buffer (pH 12) and incubated for 3 h at 60 °C. Finally, pure-protein-free [^32^P]poly(ADP-ribose) was extracted three times with isoamyl alcohol/chloroform (1:24, *v*/*v*) [72] and was analyzed by both electrophoresis on 20% polyacrylamide gel [73] and thin-layer chromatography (TLC) on PEI cellulose plates in 0.05 M ammonium bicarbonate [74]. Autoradiographic patterns of labeled poly(ADPR) on dried gel and TLC were acquired by a Phosphor-imager (mod. Fx, BioRad).

### 4.7. Poly(ADP-ribose) Glycohydrolase Activity

Two aliquots of all brain homogenates (20 μg of proteins) were incubated in a reaction mixture (final volume 50 μL) containing 0.5 M Tris-HCl (pH 7.5), 50 mM MgCl_2_, 10 mM DTT, and 0.4 mM [^32^P]NAD^+^ (10,000 cpm/nmole), in PARP assay conditions, to produce poly(ADPR) [47]. After incubation for 15 min at 25 °C, the reaction was stopped on ice by adding 20% (*w*/*v*) trichloroacetic acid (TCA). An aliquot was washed with 7% TCA and filtered on Millipore filters (HAWPP0001, 0.45 μm).

The second was centrifuged at 10,000 rpm for 5 min at 4 °C. After washing in absolute ethanol, the precipitate, consisting of poly-ADPribosylated and non-ADPribosylated proteins, was incubated in a reaction mixture (100 mM Tris-HCl (pH 8), 10 mM dithiothreitol) in the presence of homogenate (80 µg) for 15 min at 37 °C in PARG assay conditions [66]. The reaction was blocked on ice, and the precipitates obtained by adding 20% TCA were washed with 7% TCA and filtered on Millipore filters (HAWPP0001, 0.45 μm).

PARP and PARG activities were measured as acid-insoluble radioactivity by liquid scintillation in a Beckman counter (model LS 1701) and expressed as mU/mg.

### 4.8. PAGE and TLC of Poly(ADP-ribose) Degradation

An aliquot of extracted [^32^P]poly(ADP-ribose) (2000 cpm) was incubated in a reaction mixture (final volume 50 µL) containing 100 mM Tris-HCl (pH 8) and 10 mM dithiothreitol in the presence of homogenate (80 µg) for 15 min at 37 °C in PARG assay conditions [66]. The reaction was blocked on ice, and the precipitable TCA fraction was washed with absolute ethanol and subsequently analyzed by both electrophoresis on 20% polyacrylamide gel and PEI cellulose plates, as described above.

### 4.9. Statistical Analysis

Statistically significant differences were assessed by one-way analysis of variance (ANOVA), followed by Holm–Sidak’s multiple comparisons test using the GraphPad Prism 8.0.1 Software. The results were reported in the graph as the mean ± standard deviation (SD), and the minimum level of acceptable significance was *p* < 0.05. Similar letters indicate no significant difference between the densitometric analysis values of the 113 kDa band recognized with anti-PARP1 N20 antibodies and those of 68 and 87 kDa detected with anti-PARG immunoblotting.

## 5. Conclusions

For the first time, a complete system of poly(ADPribosyl)ation in the brain of zebrafish exposed to aluminum for several days was characterized. Exposure to aluminum induces genotoxic damage, resulting in increased PARP activity and PAR synthesis at 10 and 15 days of exposure. PAR turnover is ensured by PARG enzymes, which show the highest activity at the same exposure times. In the light of this evidence, we could hypothesize that the balance between PAR synthesis and degradation prevents cell death by parthanatos. On the other hand, the reduction of PARP activity and PAR synthesis associated with a decrease in PARG activity could indicate an adaptation to aluminum exposure at longer times (20 days). It is likely that neuronal cells adopt the stratagem of reducing polymer synthesis to avoid energy expenditure and allow cell survival.

## Figures and Tables

**Figure 1 ijms-24-08766-f001:**
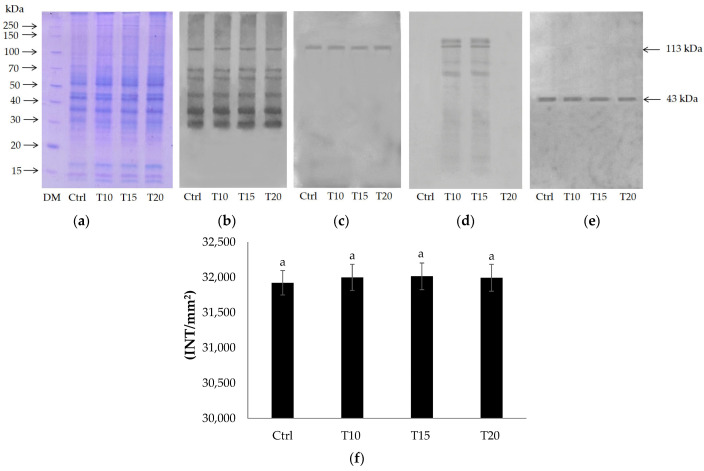
12% SDS-PAGE (**a**) and immunoblotting with anti-PARP (**b**), anti-PARP1 N20 (**c**), anti-PAR (**d**), and anti-β-actin (**e**) on control and T10, T15, and T20 samples. Densitometric analysis (**f**). Bars represent mean ± SD. Similar letters indicate no significant differences between treated groups.

**Figure 2 ijms-24-08766-f002:**
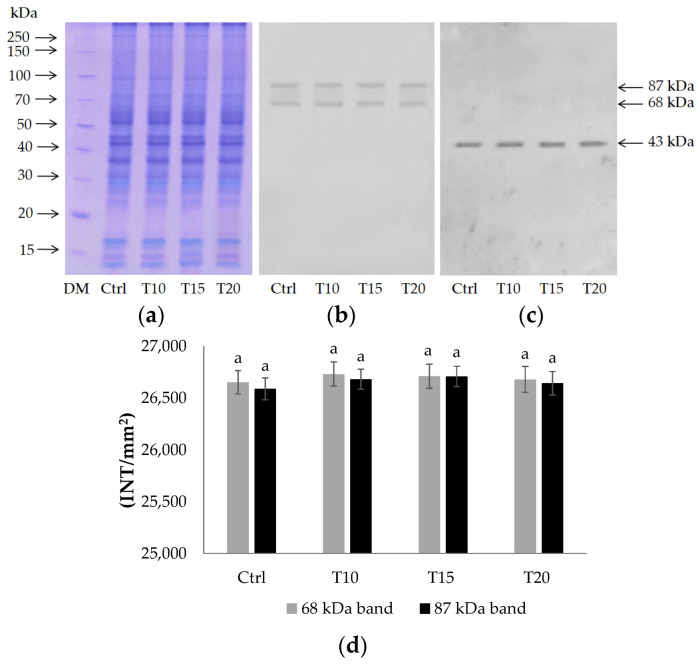
12% SDS-PAGE (**a**) and immunoblotting with anti-PARG antibodies (**b**) and anti-β-actin (**c**) on Ctrl and T10, T15, and T20 brain homogenates. Densitometric analysis (**d**). Bars represent mean ± SD. Similar letters indicate no significant differences in densitometry values between the treated groups for both immunopositive bands.

**Figure 3 ijms-24-08766-f003:**
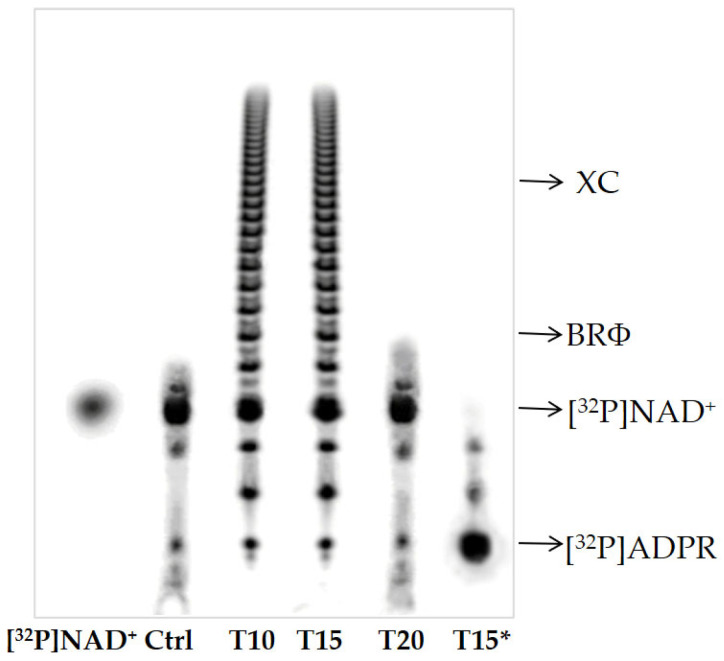
Isolation of protein-free [^32^P]-labeled poly(ADP-ribose). [^32^P]ADPR oligomers from Ctrl and T20 brain homogenates; [^32^P]ADPR polymers from T10 and T15 brain homogenates; degradation of [^32^P]poly(ADPR) produced in brain homogenates exposed to Al for 15 days (T15*). [^32^P]NAD^+^ (200 cpm), bromophenol blue (BRΦ), and xylene cyanol (XC) migrating as a tetramer, an octamer, and a 20 mer of ADP-ribose (ADPR), respectively.

**Figure 4 ijms-24-08766-f004:**
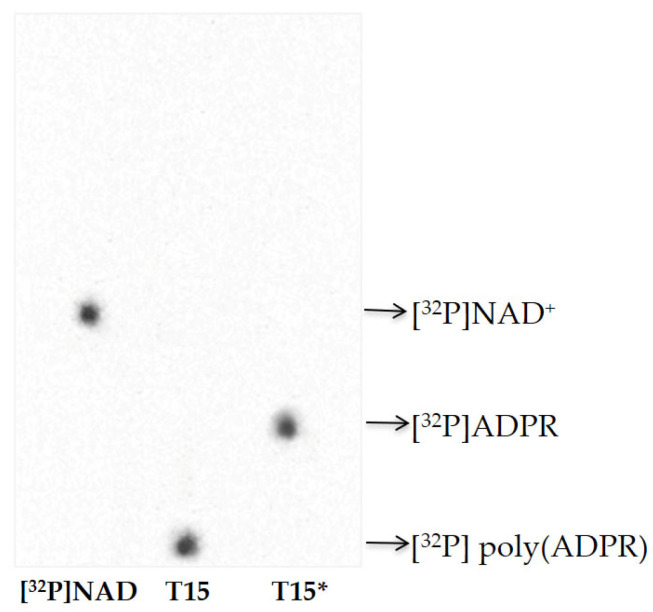
TLC of reaction products. Protein-free [^32^P]PAR (300 cpm) purified from brain homogenate exposed to Al for 15 days (T15) and degraded by PARG activity (T15*). Nucleotide standards were [^32^P]NAD^+^ (300 cpm), [^32^P]PAR (300 cpm), and [^32^P]ADP-ribose (300 cpm).

**Table 1 ijms-24-08766-t001:** Poly(ADPR) degradation percentage.

Zebrafish Brain	PARP Activity (mU/mg)	PARG Activity (mU/mg)	Poly(ADPR) Degradation (%)
Control group	53.3 ± 4.88 ^a^	0.6 ± 0.02 ^a^	99
Exposed to Al for 10 days	1086 ± 27.6 ^b^	48.3 ± 4.27 ^b^	96
Exposed to Al for 15 days	1071 ± 25.4 ^b^	24.4 ± 1.56 ^c^	98
Exposed to Al for 20 days	56.5 ± 10.69 ^a^	1.5 ± 0.36 ^a^	97

Activity values are reported as mean ± SD. Data followed by different letters in the same column are significantly different for *p* < 0.05.

## Data Availability

Data are available from the corresponding author upon reasonable request.

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
