# Peer review of "Synthesis and Degradation of Poly(ADP-ribose) in Zebrafish Brain Exposed to Aluminum"

_ijms, 2023, doi:10.3390/ijms24108766_

Round 1

Reviewer 1 Report

Manuscript title: Poly (ADPribosyl) ation characterization in Zebrafish brain exposed to Aluminum

In this manuscript, the authors characterized the complete poly (ADPribosyl) ation system in the brain of adult zebrafish exposed to 11mg/L of Aluminum for 10, 15, and 20 days. This study moved the authors’ previous study forward and gave us a comprehensive understanding of the mechanism of how Al damages the brain of the zebrafish and affects their behavior. However, I have a few comments that need further attention prior to resubmission.

1. In introduction and discussion, there are many paragraphs containing only one sentence. From the context, I think you could merge some sentences into one paragraph.

2. For the western blot data in figure 1 (b, c, and d) and figure 2 (b), do you use any reference gene as control?

Author Response

Reviewer 1 In this manuscript, the authors characterized the complete poly(ADPribosyl)ation system in the brain of adult zebrafish exposed to 11mg/L of Aluminum for 10, 15, and 20 days. This study moved the authors’ previous study forward and gave us a comprehensive understanding of the mechanism of how Al damages the brain of the zebrafish and affects their behavior. However, I have a few comments that need further attention prior to resubmission.

1 C: In the introduction and discussion, there are many paragraphs containing only one sentence. From the context, I think you could merge some sentences into one paragraph.

R: The authors are grateful for the suggestion and have merged some sentences into a single paragraph (in red) both in the introduction and discussion.

2 C: For the western blot data in Figure 1 (b, c, and d) and Figure 2 (b), do you use any reference gene as control?

R: The authors had already performed western blotting analyses with anti-actin antibodies and apologise for not showing the results. Now they have included western blots normalized to beta-actin (Figures 1e and 2c respectively) in the Results section. The procedure has been described in Materials and Methods.

Reviewer 2 Report

I have received the research article entitle “Poly(ADPribosyl)ation characterization in Zebrafish brain exposed to Aluminum” by Anna Rita Bianchi et al. for evaluation. 

In this study, authors have shown that application of Aluminum induced genotoxic damage and further biochemical alterations in Zebrafish. The study is important and helpful to understand the basic enzymatic activities responsible for the synthesis of ADP-ribose polymers (PAR) from NAD+. However, Authors have extrapolated many this without proving the facts experimentally.

I have some critics to improve the quality of this manuscript. 

My specific comments are:

1-    Author should describe in the method section how they have chosen the toxic dose (11 mg/L) of Aluminum? 

2-    How have authors concluded (experimentally) that Aluminum induced the genotoxic damage in the brain of Zebrafish in this study? 

3-    In the conclusion section, authors have summarized this study stating that Aluminum exposure also alters the survival of neuronal cells by increasing the increased synthesis of poly (ADPR). I would suggest authors to test neuronal cell survival in this study. 

4-    Authors have not performed statistical analysis for the dated presented in table 1. Why?

5-    The method section specially the treatment and doses must be improved and well explained.

6-    The title of the paper must be re-phrased. 

7-    In the abstract, authors have mentioned “Our previous study demonstrated that Aluminum (Al) alters the zebrafish cerebral tissue and actives PARP.” In line 14-15 is not clear. Please rephrase it. 

8-    I would suggest authors to provide the graphical abstract. Graphical abstract may be helpful to draw the attraction of readers. 

9-    I would suggest authors to improve the English language of the manuscript thoroughly.

The sentences are poorly phrased it is very difficult to understand/incomprehensible. Authors may consult english language expert to improve the manuscript.

Author Response

Reviewer 2 In this study, authors have shown that application of Aluminum induced genotoxic damage and further biochemical alterations in Zebrafish. The study is important and helpful to understand the basic enzymatic activities responsible for the synthesis of ADP-ribose polymers (PAR) from NAD+. However, the Authors have extrapolated many this without proving the facts experimentally.  I have some critics to improve the quality of this manuscript. My specific comments are:

1 C: Author should describe in the method section how they have chosen the toxic dose (11 mg/L) of Aluminum?  

R: The authors thank you for all the useful suggestions that have been accepted. The variations are all reported in the manuscript with the color redIn the Introduction and in Materials and Methods we have inserted the motive of this choice.

2 C: How have authors concluded (experimentally) that Aluminum induced the genotoxic damage in the brain of Zebrafish in this study?  

R: The authors have explained more clearly in the text (from 227 to 231 lines) that, being PARP a sensor of DNA damage, the increase in both PARP activity and PAR synthesis represents two indirect experimental evidence of genotoxic damage induced in Zebrafish brain exposed to Al for short times. 

3 C: In the conclusion section, authors have summarized this study stating that Aluminum exposure also alters the survival of neuronal cells by increasing the increased synthesis of poly (ADPR). I would suggest authors to test neuronal cell survival in this study.

R: Our conclusions are relative to these new biochemical data on poly(ADPribosyl)ation system that are subsequent to a previous study (https://doi.org/10.1016/j.chemosphere.2022.135752). New data confirm the previously reported trend with the evident histological alterations at 10 and 15 days of Al exposure and in according also their decrease at 20 days.

4 C: Authors have not performed statistical analysis for the dated presented in table 1. Why? 

R: The authors apologize for not reporting the statistical analysis in Table 1 and have included it. 

5 C: The method section especially the treatment and doses must be improved and well explained. 

R: We have explained better the treatment and doses in materials and methods.

6 C:  The title of the paper must be re-phrased.  

R: “Synthesis and degradation of poly(ADP-ribose) in Zebrafish brain exposed to Aluminum” is the new title.

7 C:  In the abstract, authors have mentioned “Our previous study demonstrated that Aluminum (Al) alters the zebrafish cerebral tissue and actives PARP.” In line 14-15 is not clear. Please rephrase it.  

R: In the abstract, the clarification is marked in red (lines 13-16).  

8 C:  I would suggest authors to provide the graphical abstract. Graphical abstract may be helpful to draw the attraction of readers.  

R: A Graphical abstract has been attached.

9 C:  I would suggest authors to improve the English language of the manuscript thoroughly.

R: The manuscript has been totally revised by an expert in the English language.

Round 2

Reviewer 2 Report

Congratulation